# Interfacial Compatibilization into PLA/Mg Composites for Improved In Vitro Bioactivity and Stem Cell Adhesion

**DOI:** 10.3390/molecules26195944

**Published:** 2021-09-30

**Authors:** Meriam Ben Abdeljawad, Xavier Carette, Chiara Argentati, Sabata Martino, Maurice-François Gonon, Jérémy Odent, Francesco Morena, Rosica Mincheva, Jean-Marie Raquez

**Affiliations:** 1Laboratory of Polymeric and Composite Materials (LPCM), Center of Innovation and Research in Materials and Polymers (CIRMAP), University of Mons, 23 Place du Parc, 7000 Mons, Belgium; Meriam.benabdeljawad@umons.ac.be (M.B.A.); xa92@hotmail.com (X.C.); jeremy.odent@umons.ac.be (J.O.); 2Department of Chemistry, Biology and Biotechnology, University of Perugia, 06123 Perugia, Italy; chiara.argentati@unipg.it (C.A.); sabata.martino@unipg.it (S.M.); francesco.morena@unipg.it (F.M.); 3Department of Material Sciences, University of Mons, 20 Place du Parc, 7000 Mons, Belgium; mauricefrancois.gonon@umons.ac.be

**Keywords:** interfacial adhesion, amphiphilic copolymer, bioactivity, stem cells, biomaterial interaction and bone implant

## Abstract

The present work highlights the crucial role of the interfacial compatibilization on the design of polylactic acid (PLA)/Magnesium (Mg) composites for bone regeneration applications. In this regard, an amphiphilic poly(ethylene oxide-b-L,L-lactide) diblock copolymer with predefined composition was synthesised and used as a new interface to provide physical interactions between the metallic filler and the biopolymer matrix. This strategy allowed (i) overcoming the PLA/Mg interfacial adhesion weakness and (ii) modulating the composite hydrophilicity, bioactivity and biological behaviour. First, a full study of the influence of the copolymer incorporation on the morphological, wettability, thermal, thermo-mechanical and mechanical properties of PLA/Mg was investigated. Subsequently, the bioactivity was assessed during an in vitro degradation in simulated body fluid (SBF). Finally, biological studies with stem cells were carried out. The results showed an increase of the interfacial adhesion by the formation of a new interphase between the hydrophobic PLA matrix and the hydrophilic Mg filler. This interface stabilization was confirmed by a decrease in the damping factor (tanδ) following the copolymer addition. The latter also proves the beneficial effect of the composite hydrophilicity by selective surface localization of the hydrophilic PEO leading to a significant increase in the protein adsorption. Furthermore, hydroxyapatite was formed in bulk after 8 weeks of immersion in the SBF, suggesting that the bioactivity will be noticeably improved by the addition of the diblock copolymer. This ceramic could react as a natural bonding junction between the designed implant and the fractured bone during osteoregeneration. On the other hand, a slight decrease of the composite mechanical performances was noted.

## 1. Introduction

The choice of the bioresorbable material is one of the crucial factors for the design of novel bone implants. The latter must fulfil additional requirements compared to permanent implants available on the market, e.g., to be metabolized by the human body without leaving any trace and gradually lose their mechanical strength during the healing process until bone regeneration [1]. More particularly, they must be designed to degrade at a rate that will slowly transfer load from the implant to the healing bone [2]. In this regard, several polymers have appeared as potential candidates to be investigated as biodegradable implants. Amongst all, poly(α-hydroxy acids) family derived from lactide and glycolide, and (ε-caprolactone) are the most commonly ones [3,4,5]. For this reason, due to its biodegradability, biocompatibility, stiffness and thermoplastic processability, PLA is widely used [6,7]. However, PLA devices suffer from insufficient interactions with bone tissue because of the lack of affinity between the hydrophobic polymer and the hydrophilic bone [8]. Moreover, inflammatory response of acid byproducts issued from PLA degradation in the late stage of implantation [8] and poor osteoconductivity limit PLA further applications in bone healing [9]. The PLA limitation in osteosynthesis implants is also associated with its hydrophobic surface, which is responsible for its limited cell adhesion [8], as well as its stiffness inferior to the values of some long bones such as cortical bone (E = 10–30 GPa) [10]. To counterbalance the aforementioned issues, many efforts have been addressed by loading the polymer matrix with hydroxyapatite filler [11,12], hydroxyapatite combined with graphene [13], calcium phosphate [14], bioactive glass ceramics [15] and bioactive fillers such as calcium silicates (CaSi) and dicalcium phosphate dihydrate (DCPD) [16] for bone tissue engineering, bone graft substitute and dentistry applications. 

Nowadays, several studies have been performed to overcome the limitations of PLA for bone regeneration. Indeed, Gandolfi et al. demonstrated that PLA-based mineral-doped scaffolds present high porosity and differentiation of cultured hPCy-MSCs for regenerative healing in dentistry [17]. The same group proved that highly porous CaSi-DCPD doped poly (α-hydroxy) acids scaffolds were able to release biologically relevant ions and nucleate apatite on the surface [16]. Interesting results were also obtained using mineral-doped poly(L-lactide) acid scaffolds enriched with exosomes [18].

Another promising biodegradable material for bone regeneration developed during the last decade is magnesium (Mg) [19,20]. Indeed, Mg content in the human body is mainly concentrated in the bone (60%–70%) [21]. Moreover, it provides excellent bioresorbability, biodegradability and more similar mechanical properties to those of bones compared to any other metal or polymer [22]. Mg-based orthopaedic implants have presented attractive abilities of bioactive behaviour, osteoconductivity and osseointegration, and have encouraged bone growth [23]. Mg inorganic filler was also embedded in a PLA and was found not only to enhance its mechanical performances but also to increase the bioactivity and so the improvement of protein adsorption leading to more cell adhesion and growth [22,23].

However, despite this good choice of biomaterials, the surface properties between the polymeric matrices and the inorganic filler constrained the successful design of bioresorbable implant as it greatly impact the interactions between the implant and the biological medium. Luque-Agudo et al. has applied surface tension calculations to understand the quality of the interface interactions between Mg microparticles and PLA matrix and their impact on bacterial adhesion and viability [24]. The lack of affinity between the hydrophobic PLA matrix and the hydrophilic Mg microparticles have been highlighted and stated to present a crucial issue in the PLA/Mg design. Cifuentes et al. visualized the low interface adhesion between PLA and Mg through morphological analysis. In order to overcome this issue and so to increase the mechanical properties, the use of coupling agents is proposed [25]. Additionally, Ferrandez-Montero et al. improved PLA/Mg interface through surface modification of Mg particles by two different stabilizers [26]. Despite the undoubted improvement, all three methods did not satisfy all requirements for successful PLA/Mg bone-regeneration materials design.

Therefore, and inspired by the literature [27,28], the present study proposes the use of an amphiphilic poly(ethylene oxide-b-L,L-lactide) diblock copolymer (PEO-*b*-PLLA) to control the interface interactions between PLA and Mg. Indeed, Raquez et al. already discussed the fact that the PEO-b-PLLA copolymer blocks enhance the biocompatibility and the hydrophilicity of PLLA/pseudowollastonite composites [28]. This last feature represents a significant factor to improve the formed hydroxyapatite layer/composite surface adhesion during the in vitro degradation test in SBF and therefore the material bioactivity. In this research, it has been proven that the more the surface is hydrophilic, the more the hydrophilic bone-like layer adheres [28]. The hydrophilicity produced by the PEO-b-PLLA copolymer is also reported in Hongji et al. research [29]. 

To the best of our knowledge, no investigation about the improvement of PLA/Mg interface by using this kind of amphiphilic surfactant has been highlighted yet. The latter is able to interact with the Mg filler via its hydrophilic part and to the PLA matrix via its hydrophobic part. In addition to their effect on the PLA/Mg biocompatibility, the amphiphilic copolymer is also able to enhance the composite hydrophilicity by selective surface localization of the hydrophilic PEO. Furthermore, it has a beneficial biological effect on the composite materials, i.e., the protein adsorption and the cell proliferation.

In order to accomplish their function, the novel implants have to combine excellent bioactivity and long-term mechanical endurance. Therefore, this paper first conducted a full-scale study on the morphology, wettability behaviour, thermal, thermo-mechanical and mechanical properties, subsequently, the bioactivity behaviour during the biodegradation process in the SBF and finally biological activity in the presence of the amphiphilic copolymer.

## 2. Results and Discussion

### 2.1. Dispersion Test

In order to study the influence of the copolymer on Mg filler dispersion (amount of 5, 10 and 15 wt.%), 2 sets of chloroform solutions were prepared, i.e., in the absence of copolymer and with 10 wt.% of PEO-*b*-PLLA block copolymer, respectively. As it can be seen in the absence of copolymer (Figure 1A), the Mg fillers are unstable in the chloroform and settled down directly because of their hydrophilic nature. In contrast, even after 4 h (which are sufficient to prepare a masterbatch with well dispersed fillers), a good dispersion of the microparticles was still observed in the presence of 10 wt.% of PEO-*b*-PLLA (Figure 1B). The use of the block copolymer has already been highlighted in Meyer et al. work [30]. Our data reveal an improved filler stabilization at 10 wt.% of copolymer content and therefore, microcomposites based on PLA, Mg filler (5, 10 or 15 wt.%) and 10 wt.% of copolymer were produced using extrusion technology (Table 1).

### 2.2. Morphology Observations

The morphology of all composites was visualized using SEM to evaluate the influence of PEO-*b*-PLLA copolymer on the PLA/Mg interfacial adhesion. The latter is a crucial parameter for good mechanical properties of the final implant. In the absence of PEO-*b*-PLLA copolymer (Figure 2A), the presence of holes around the microparticles and the cavities left by the Mg after cryofracture attests for the lack of interactions between the hydrophobic PLA matrix and hydrophilic Mg particles. This behaviour is considered typical for incompatible composites with low interfacial adhesion between the matrix and the filler [25,31]. In contrast, the presence of the amphiphilic copolymer allowed interestingly good compatibility between the matrix (thanks to its hydrophobic part) and the Mg filler (thanks to its hydrophilic part) to form a new interphase as showed in Figure 2B by the absence of holes and cavities. In addition to that, the surface is smoother and more homogenous after cryofracture coming from the stabilized interface that can better respond to the stress during cryofracture.

### 2.3. Water Contact Angle

After confirming the positive effect of the copolymer on the adhesion between the micro-filler and the matrix, the effect of the amphiphilic copolymer on the composites surface properties was studied upon WCA measurements. Indeed, studies on PLA blends with PEO (co)polymers show significant reduction in material’s contact angle [32]. Figure 3 shows that the PLA has a contact angle of 93° in good agreement with literature data [23]. The addition of 5, 10 and 15 wt.% of the hydrophilic Mg filler renders the composite surface less hydrophobic with a contact angle of 84, 81 and 79°, respectively, as expected. In fact, Coronada Fernández-Calderón et al. measured highly hydrophobic water contact angle of PLA (95° ± 1°) which decreased to 72° ± 1° after addition of Mg fillers [23]. Again, the PEO-*b*-PLLA block copolymer presence shows a positive effect on the composite wettability due to the hydrophilic nature of the PEO blocks. In other words, for a given amount of Mg, the presence of 10 wt.% block-copolymer further reduced the water contact angle at approximately 10%. The result was considered as being encouraging since the materials wettability is essential for cell adhesion, the formation of the apatite layer and therefore the consecutive tissue regeneration [27,33]. 

### 2.4. Thermomechanical Properties 

DMTA results of all the composites are given in Appendix A. The storage modulus or elastic modulus (E′), which can be considered as a direct measurement of the stiffness of the material and the tan δ (E″/E′) that is useful for evaluating the molecular chain mobility as well as the quality of interfacial adhesion between filler and polymer chains, were reported at 37 °C. Regarding the PLA/x Mg set, an improvement of E′ was observed within the investigated temperature. In fact, the value of the storage modulus increases up to 10 wt.% of Mg filler then falls slightly. This indicates that the inorganic load has an interesting effect on the PLA matrix stiffness. According to the literature, this could be due to the well dispersion of the filler within the PLA [34]. The decrease at 15 wt.% could be explained by the presence of some filler agglomerates in PLA matrix which is in agreement with the SEM images (Figure 2A). The T_tan δ_ (°C) of the composites were determined from the temperature values corresponding to the maximum of tan δ curves. The presence of Mg particles leads to a slight shift of the tanδ curves towards the highest temperatures at weight ratio of 5 and 10 % compared to that of the neat PLA (Figure 4A). This can be associated to the chain mobility restriction due to the presence of fillers [8]. However, a decrease in the T_tanδ_ (°C) was reported at 15 wt.%. in filler. This behaviour has been ascribed to some defects introduced to the PLA matrix after the addition of the inorganic filler. According to the literature, the T_tanδ_ (°C) decrease could be restrained by the PLA/Mg interface improvement [35]. Concerning the PLA/10 Copo/x Mg set (Figure 4B), for a similar inorganic load in the composites, the storage moduli reproduce the same behaviour of the first set with a decrease in the values. In reality, the E′ decrease indicates the plasticizing effect of the PEO present in the PEO-*b*-PLLA copolymer, which is well dispersed in the PLA matrix [36]. This behaviour is accompanied with a slight decrease in T_tanδ_ (°C) of the PLA/10 Copo, which is correlated with the DSC results (Appendix A). Finally, a significant loss of T_tanδ_ (°C) is detected when Mg amount increases. Comparing the Figure 4A and Figure 4B, it is worth noting that the fact of the tan δ peak width at half height be broader for the second set (Figure 4B) is an indicative of better PLA/Mg adhesion [37]. Furthermore, Ko et al. also reported a decrease of tan δ with the highest intensity of interface adhesion [38]. Consequently, the composites reinforced with an amount of 10 wt.% of Mg filler will be more investigated in the following section as they present the highest mechanical reinforcement in both cases.

### 2.5. Mechanical Properties

The compressive modulus and strength at yield (Figure 5B) were calculated from the stress–strain curves (Figure 5A). First, it has been noticed that all the curves showed the same behaviour and exhibit several characteristic stages. Indeed, initially, the samples show a reversible viscoelastic deformation, where stress grows linearly with strain, so the compressive moduli are determined. Subsequently, the deformation becomes irreversible, which is called the yield point [39]. Then, the stress decreases, which is recognizable by the strain softening. Finally, the stress increases again with further deformation, which is referred to the strain hardness. According to Figure 5B a decrease of 8% of the compressive modulus of PLA was noted following the addition of PEO-b-PLLA copolymer, which can be attributed to the plasticizing nature of the PEO [40]. The introduce of Mg microparticles within the PLA led to an increase of 30 % of the PLA modulus accompanied with an increase of 23 % of the compressive strength at yield, which is in accordance with the DMTA results. The PLA/10 Copo/10 Mg composite showed an increase of 16% of the compressive modulus compared to the PLA/10 Copo. This result confirms our previous discussion about the DMTA and SEM analyses (section before) that reveals the improvement of the interfacial adhesion in the presence of the copolymer.

### 2.6. Hydrolytic Degradation Behaviour 

#### 2.6.1. Macroscopic Observations

The macroscopic aspect of the processed composites and the neat PLA is presented in Figure 6 at first sight during the degradation in SBF. The neat PLA and PLA/10 Copo specimens showed a significant change of the opacity noted already after 2 weeks of immersion for the PLA/10 Copo and 6 weeks for the neat PLA. Increasing the hydrolysis time led to an increase in relative opacity for both. In this regard, previous papers were discussed the opacity modification of PLA during the hydrolysis in buffered medium [40,41]. Indeed, this change of the opacity can be explained by various phenomena such as (i) the light scattering due to the absorbed water through the material, (ii) the formation of holes and then cracks on the surface as well as in the inner part of the specimens, (iii) the degradation products formed during the hydrolysis and (iv) the increase in PLA crystallinity [42,43]. It is worth noting that the opacity is more pronounced in the presence of copolymer. Concerning the reinforced composites, a prominent change of colour is seen, i.e., gradually from grey to white colour. In fact, a white layer was formed on the specimens surfaces [25]. Its thickness was increased over time with both the addition of Mg fillers (5, 10 and 15 wt.%) and the presence of PEO-*b*-PLLA copolymer. According to the literature, the hydrolytic degradation is initiated when the water diffused into the matrix. Throughout the PLA, water molecule was reached the Mg surface and formed a white layer that mainly consist of magnesium hydroxide (Equation (1)) [20,26].
(1)$$\mathrm{Mg}+2{\;H}_{2}O\;\to {\;\mathrm{Mg}}^{2+}+2{\mathrm{OH}}^{-}+H_{2}\left(g\right)\to \mathrm{Mg}{\left(\mathrm{OH}\right)}_{2}+{\;H}_{2}\left(g\right)$$

The difference of colour change is obvious comparing the two sets of composites, which is deeply explained by the characterizations in following sections. Additionally, during the immersion time, some samples became brittle because of the formation of Mg(OH)_2_ [25]. Moreover, from the fourth week, some cracks were detected, especially in the PLA/10 Copo/x Mg set. The choice of 10 wt.% of Mg filler is confirmed by combining the material stiffness and the macroscopic observations. The degradation behaviour of the composites (molecular weight, crystallinity, mass and pH evolution) is presented in Appendix A.

#### 2.6.2. Ion Release Measurements 

The change of ions (Mg, Ca and P) concentration in the SBF medium of PLA/10 Mg and PLA/10 Copo/10 Mg composites was studied during the degradation process. The concentration of Mg increased from the 6th week Figure 7A and from the 4th week Figure 7B following the addition of copolymer. Simultaneously, the concentration of Ca and P decreased slightly by a similar trend (Appendix A) for both composites, which indicates that the corrosion of Mg promotes Ca and P deposition, which is the first proof of the osteoconductive properties of our materials. This behaviour has already been highlighted in the literature by Jing Bai et al. [19] and H. Cai et al. [44].The corrosion is more important in the presence of the copolymer which is in agreement which the hydrogen release (Appendix A) and EDX (next section) results.

#### 2.6.3. Composite Morphology and Characterization of Degradation Products 

In order to assess the morphological evolution on the surface in absence of copolymer and after its incorporation in the composite, SEM was performed on samples during the degradation process (after 2, 4, 6 and 8 weeks, respectively) as shown in Figure 8A,B. The images after 10 and 12 weeks are not presented because of the manipulation difficulty due to the appearance of cracks. Indeed, according to Figure 8A some corrosion pits can be observed because of the significant degradation of Mg filler over time [45]. Jiao et al. reported a similar trend of morphology with the pure Mg [46]. After 8 weeks of immersion, a few slight cracks were appeared on the composite surface. As far as Figure 8B, the presence of pitting corrosion was more important which develop into large cracks after 4 weeks. This behaviour was expected due to the raise of degradation following the addition of the amphiphilic copolymer. Combined with EDX results (Table 2), the predominant components formed on the surface were C, O, Ca and P elements, in addition to trace of Mg. Indeed, the content of Mg was decreased gradually, in both cases, with the immersion time extension (4, 6 and 8 weeks). The degradation of Mg promotes the deposition of P and Ca which will be beneficial for new bone tissue formation during the healing process [44]. Subsequently, the Ca/P weight ratio of the degradation layer was calculated. After 8 weeks of soaking in SBF, 1.67 was reached for the PLA/10 Mg (Table 2A) versus 1.00 for the PLA/10 Copo/10 Mg (Table 2B). Thus, a layer of hydroxyapatite was formed on PLA/10 Mg surface. Its absence following the addition of copolymer could be explained by the formation of large cracks therefore the instability of the surface. For this raison, cross-sections were also analysed to identify the chemical composition in bulk as it is presented in Figure 9. Interestingly, the content of Ca and P was more prominent in the case of PLA/10 Copo/10 Mg (Figure 9B and Table 3B). Indeed, 1.67 of Ca/P ratio was reached after 8 weeks of degradation versus 1.00 for the PLA/10 Mg (Figure 9A and Table 3A).

The EDX results were also supported by the XRD patterns as shown in Figure 10. The diffractogram of PLA/10 Mg in Figure 10A before the immersion time (PLA/10 Mg-0w) gave rise to the three strong reflections at 32.2°, 34.4° and 36.6°, which correspond respectively to the (100), (002) and (101) Mg planes [39] and other small reflections which also correspond to (102), (110), (103) and (112) Mg planes, respectively [23]. The domination of Mg peaks decreased over time and the components of corrosion products were formed and further identified as major Mg(OH)_2_ and a small number of (Ca)_3_(PO_4_)_2_ and (Mg)_3_(PO_4_)_2_. According to the literature, (Ca, Mg)_3_(PO_4_)_2_ has similar chemical compositions as the nature bone which could induce good biocompatibility and osteoconductivity [47,48]. During the degradation process, H_2_PO_4_^−^ dissolved in the SBF bath and reacted with OH^−^ from Equation (1) to produced HPO_4_^2−^ and PO_4_^3−^ respectively (Equation (2) and Equation (3)).
(2)$$H_{2}{\mathrm{PO}}_{4}^{-}+{\;\mathrm{OH}}^{-}\to {\;\mathrm{HPO}}_{4}^{2-}+{\;H}_{2}O$$
(3)$$H_{2}{\mathrm{PO}}_{4}^{-}+2{\;\mathrm{OH}}^{-}\;\to {\;\mathrm{PO}}_{4}^{3-}+2{\;H}_{2}O$$

Then, Ca^2+^ and Mg^2+^reacted with HPO_4_^2−^ and PO_4_^3−^ to form small quantities of Ca_3_(PO_4_)_2_ and Mg_3_(PO_4_)_2_, respectively (Equation (4) and Equation (5)) [49,50].
(4)$$3{\;\mathrm{Ca}}^{2+}+2{\;\mathrm{PO}}_{4}^{3-}\;\to {\;\mathrm{Ca}}_{3}{\left({\mathrm{PO}}_{4}\right)}_{2}$$
(5)$$3{\;\mathrm{Mg}}^{2+}+2{\;\mathrm{PO}}_{4}^{3-}\;\to {\;\mathrm{Mg}}_{3}{\left({\mathrm{PO}}_{4}\right)}_{2}$$

Regarding Figure 10B, two peaks located at 2θ = 16.54° and 18.83° were recorded, which indicate the semi crystallinity of the PLA [51]. The more the peaks increased over time, the more prominent the crystallinity is. This behaviour has been already confirmed by DSC analyses (Appendix A) The appearance of the first peak at 16.54° before the immersion in SBF could be explained by the role performed by the Mg particles as a nucleating agent and the PEO as a plasticizer in the matrix which can accelerate the nucleation and its crystal growth of PLA [52]. In this case, the major corrosion product deposed on the PLA/10 Copo/10 Mg surface was the Mg(OH)_2_ which effectively confirms the EDX results. Indeed, its Ca/P ratio was very low (Ca/P = 1.00) comparing to the PLA/10 Mg composite (Ca/P = 1.67).

#### 2.6.4. Mechanical Characterization 

Compression test was performed every 4 weeks on PLA, PLA/10 Copo, PLA/10 Mg and PLA/10 Copo/10 Mg during the degradation process. The compressive moduli (Figure 11A) and strength at yield (Figure 11B) were calculated from the stress–strain curves. After one month of degradation, all the mechanical performances were decreased. Thereafter, the loss became grater, except the pure PLA which maintained its mechanical properties during the whole test. This behaviour could be explained by the increase of the crystallinity over time especially with the slight decrease in the molecular weight and the mass variation compared with the other samples (Appendix A). Regarding the PLA/10 Mg, this implant meets the compressive strength of natural cortical bone during the first 4 weeks. According to the literature, this value is between 100 and 230 MPa [53,54,55]. On the other hand, the compressive Moduli of all the materials until the 8th week are within the range of cancellous bone (50–800 MPa) [56]. This behaviour has already been highlighted in the literature by Ali et al. [20]. Mg content of 10 wt.% improves the initial mechanical properties of PLA which was proved by Cifuentes work [25]. However, it impairs both compressive modulus and strength at yield during the degradation process, which is further accelerated with the presence of the amphiphilic PEO-*b*-PLLA copolymer. Indeed, the decrease in the mechanical performances with the in vitro degradation time can be explained by the hydrolysis of PLA and also the corrosion of Mg filler which forms a Mg(OH)_2_ layer, which leads to a rise of hydrolysis at PLA/Mg interface. Therefore, there is a deterioration of the adhesion between the matrix and the filler. The copolymer enhances this behaviour as it facilitates the diffusion of water in the composites over time. 

### 2.7. Protein Adsorption 

First, we evaluated the protein-binding capability of different polymer film used. This information is necessary for guaranteeing the interaction of the cellular proteins with the film surface. The capability of PLA-based films to bind proteins was measured trough the protein adsorption assay (Figure 12). We assessed the adsorption of BSA (2 mg/mL), FBS 2%, FBS 10% and plasma (5 mg/mL), on each film at two different incubation times, 30 min and 24 h, according to our protocol [57,58]. We found an increase of the adsorption capability in PLA-derivative films compared to neat PLA film. The increase was measured for all proteins and was in the order PLA/10 Copo/10 Mg > PLA/10 Mg > PLA/10 Copo > PLA film (Figure 12). This behaviour was observable after 30 min of incubation and was further increased after 24 h of incubation. Of note, the highest increase in the adsorption capability of PLA-based films for FBS 10% and plasma (>FBS 2% and >BSA) was due to the highest protein concentration in these samples (Figure 12). The results suggested that the incorporation of the amphiphilic diblock copolymer (PEO-*b*-PLLA) increased the capacity of polymer films to bind different kinds of proteins thanks to its hydrophilicity. This behaviour is in agreement with the aforementioned WCA measurements.

### 2.8. Culture of Human Adult Mesenchymal Stem Cells on PLA-Based Films

In this work, we have also investigated the interaction of PLA, PLA/10 Copo, PLA/10 Mg and PLA/10 Copo/10 Mg with two types of human MSCs: hBM-MSCs and hASCs [59,60,61,62]. We have chosen hBM-MSCs and hASCs based on the already established protocols for in vitro culture (both expansion and differentiation toward non-hematopoietic cell lineages such as fibroblasts, osteoblasts, adipocytes, chondroblasts, and neural cells) as well as their immunomodulatory activity, which makes them suitable for regenerative medicine, including tissue engineering strategies and stem cell basic research [63,64,65,66] (see Appendix A). For these reasons MSCs provide new insights into the advancement of therapeutic strategies for different diseases (e.g., cardiovascular and neurodegenerative) [67,68] and the development of a biohybrid system for tissue engineering applications, where the cross-talk between stem cells and the surface of the biomaterial is the main event [69]. In fact, biomaterial chemical-physical cues serve as biochemical signalling, which stem cells detect and transduce in biological functions such as adhesion, proliferation, morphology and differentiation [70,71].

To investigate the biological properties of films, PLA, PLA/10 Copo, PLA/10 Mg and PLA/10 Copo/10 Mg polymer films were used as a support for culture of hBM-MSCs and hASCs. We evaluated stem cell proliferation, viability and cell morphology at different time points (3, 7 and 14 days (D)). As CTR, experiments were performed seeding both MSCs on TCP (for cell proliferation and viability assay) or GC (for the morphology analysis).

#### 2.8.1. Stem Cells Proliferation and Viability

First, we measured the proliferation rate of hBM-MSCs and hASCs on PLA, PLA/10 Copo, PLA/10 Mg and PLA/10 Copo/10 Mg polymer films (Figure 13A,B). We found a comparable growth curve of hBM-MSCs on PLA and control TCP, while we observed a significant increase in the proliferation rate on PLA/10 Mg (Figure 13A). Conversely, we found a strong reduction of the cell proliferation on PLA/10 Copo that was restored on the same film containing Mg (PLA/10 Copo/10 Mg) (Figure 13A). Similar behaviuor was recorded for hASCs on PLA-derivative films (Figure 13B). In fact, we found a similar growth curve of hASCs on PLA and TCP, a high increase on PLA/10 Mg, a significant reduction on PLA/10 Copo and a restored on the film PLA/10 Copo/10 Mg (Figure 13A).

Together these results indicated that the presence 10 Copo in PLA interfered with the cell proliferation and that the inclusion of 10 Mg in the polymer film re-establishes the stem cell proliferation rate (Figure 13A,B).

The MTT assay was used to monitor the stem cells viability by measuring the activity of mitochondrial dehydrogenase at different time points.

Results reported in Figure 13C,D showed a comparable dehydrogenase activity in both types of MSCs on PLA and PLA/10 Copo/10 Mg compared to CTR at each time point confirming the proliferation rate (Figure 13A,B). Moreover, according to the growth curve, we found a significant increase of the enzymatic activity on PLA/10 Mg compared to CTR at D14, and a high reduction at D7 and D14 in both stem cells on PLA/10 Copo respect to CTR (Figure 13C,D). 

These findings demonstrated that PLA, PLA/10 Mg, and PLA/10 Copo/10 Mg were suitable for mesenchymal stem cell cultures, while PLA/10 Copo film has a potential adverse effect on the viability of both stem cells.

#### 2.8.2. Mesenchymal Stem Cell Shape on PLA-Based Films

To evaluate the stem cells morphology, we analysed by immunostaining the cytoskeletal F-actin fibres of hBM-MSCs and hASCs cultured on PLA-based films. The analysis was performed at days 3, 7 and 14 (Figure 14).

The data in Figure 14A,B showed that hBM-MSCs and hASCs on PLA, PLA/10 Mg and PLA/10 Copo/10 Mg films maintained a monolayer fibroblast-like morphology similar to stem cells in control condition on GC. In particular, F-actin fibres travels the whole cell, crossing the cytoplasm and orienting parallel to the main longitudinal axis of the cell. Conversely, on the PLA/10 Copo the hBM-MSCs film lost their canonical morphology, became smaller and irregular, while hASCs generated big aggregates attached to the film surface starting from D3. The immunostaining of F-actin of hBM-MSCs and hASCs also showed that cell number was significantly reduced on PLA/10 Copo with respect CTR and the others film (Figure 13 and Figure 14), confirming our previous data of stem cells viability (Figure 13). Of note, the presence of 10 wt.% Mg in PLA/10 Copo seemed to restore the canonical growth and cellular morphology of both MSCs (Figure 13 and Figure 14). The overall data, as confirmed also with the immunostaining of cytoskeletal F-actin fibres, showed that these films could serve as implants for biomedical applications. Of note, we observed an adverse effect of PLA/10 Copo on both stem cells that consequently lost the canonical morphology, shrank, and became irregular (hBM-MSCs) or formed large aggregates attached to the film surface beginning at D3 (hASCs). The canonical stem cells behaviour on PLA/10 Copo/10 Mg suggests that the presence of Mg on PLA/10 Copo was capable of reverting the adverse effect. More experiments are necessary to explain the mechanism. In conclusion, in accordance with our previous works, all these data confirmed that the two types of stem cells used exhibit a different interaction and behaviour on the same film surface [57,72].

## 3. Materials and Methods

### 3.1. Materials

L,L-Lactide (LLA) was supplied by Purac Biochem, PURASORB (Gorinchem, The Netherlands) and hot-recrystallized twice in dry toluene before use. Semi-crystalline poly(L,L-lactide), hereafter called PLA, was provided by NatureWorks LLC (US) (grade 6201D, number-average molar mass (M_n_) = 80.000 g/mol, dispersity (Ð = 2), obtained by gel permeation chromatography (GPC) calibrated with PS standards in CHCl_3_ at 30 °C, D-isomer <2% according to the supplier). Poly(ethylene glycol) monomethyl ether with a M_n_ of 5.000 g/mol (as determined by proton nuclear magnetic resonance (^1^H NMR) in CDCl_3_) with a Ð of 1.25 (as determined by GPC in CHCl_3_) and phosphate buffered saline (PBS), pH = 7.2, were provided by VWR (Oakville, ON, Germany). The simulated body fluid (SBF) was prepared using (NaCl, NaHCO_3_, Na_2_SO_4_, HCl), (KCl, NaN_3_), (MgCl_2_.6H_2_O) and (K_2_HPO_4_·3H_2_O, CaCl_2_, (CH_2_OH)_3_CNH_2_) from VWR, Across, Merck and Sigma Aldrich (Søborg, Denmark), respectively, and used as received. Spherical Mg particles of less than 50 µm in diameter, 99% of purity, supplied by Nitroparis (Castellón, Spain), were used (Appendix A). Tin octoate (Sn(Oct)_2_), from Sigma-Aldrich, was diluted in dry toluene (2.5 × 10^−2^ M). The latter was dried using an MBraun solvent purification system under N_2_. Bovine serum albumin (BSA) was purchased from Sigma Aldrich, St. Louis, MI, USA. Fetal bovine serum (FBS), DMEM High Glucose, penicillin–streptomycin, glutamine, RPMI-1640 and L-glutamine were from Euroclone S.p.A, Pero (MI), Italy. The plasma was from adult donors. DAPI (4,6-diamidino-2-phenylindole) was provided from Vector Laboratories Inc., Burlingame, CA, USA.

### 3.2. Synthesis of Poly(ethylene oxide-b-L,L-lactide) Diblock Copolymer (PEO-b-PLLA)

Ring-opening polymerization (ROP) of L-lactide proceeded with PEO (chain length of 5000 g/mol) as macroinitiator in the presence of tin(II) octoate (Sn(Oct)_2_) as catalyst was done in bulk in the reactor from Autoclave France (Appendix A) under N_2_. The reaction was carried out at an initial Sn(Oct)_2_/PEO-OH molar ratio of 0.05 during 30 min at 180 °C as previously performed in our laboratory [28]. The resulting mixture was dissolved in chloroform then precipitated in cold heptane (8 times the volume of CHCl_3_). PEO-*b*-PLLA copolymer was recovered after drying under vacuum at 60 °C overnight. The purification of the polymer was performed as referred to Raquez et al. [28]. The PEO-*b*-PLLA block copolymer was characterized: $M_{n\;\mathrm{PLLA}\;\left(\exp\right)}$= 5040 g/mol and conversion (%) = 95 as determined by ^1^H NMR and dispersity = 1.43 as determined by GPC analysis.

### 3.3. Preparation of PLA/Mg Microcomposites

PEO-*b*-PLLA/Mg masterbatches were performed as follows: PEO-*b*-PLLA copolymer 10 wt.% chloroform solutions were prepared and different amounts of Mg fillers (5, 10 and 15 wt.%) were dispersed by stirring at r.t. to form viscous suspension. A second set of chloroform solutions was carried out under the same conditions but without copolymer. The solvent was used in order to avoid the loss of Mg microparticles during the extruder feeding and increase the filler dispersion. All the solutions were dried at 60 °C under reduced pressure for 24 h to eliminate any trace of solvent. The obtained films were cut and added to PLA pellets (dried at 60 °C overnight under vacuum to minimize water content and avoid any excessive degradation upon processing). Melt-compounding was performed at 180 °C under nitrogen in a co-rotating twin-screw, DSM microcompounder, 15 cc, to obtain PLA/Mg composites. The rotation speed of the screws was 30 rpm for 5 min upon feeding, then 60 rpm for 5 min upon mixing. The different PLLA/Mg formulations were moulded by thermocompression using the following procedure: preheating at 180 °C for 3 min under low pressure, then 3 degassing cycles were performed at 3, 6 and 9 bar for 3 s, and then released for 1 s after each pressure step (to get rid of air-bubbles). Finally, the last pressure moulding was performed during 2 min under 10 bar. The samples were then cooled by compressing at 20 °C under 5 bar for 5 min. For the hydrogen release, degradation assessment and compression test, cylindrical samples of 12 mm length and 4 mm diameter were prepared. Rectangular specimens of 60 × 12 × 3 mm^3^ (length × width × thickness) for dynamic mechanical thermal analysis (DMTA) and films of 150 µm thickness for cell culture tests were moulded.

### 3.4. Hydrogen Release Assessment

The hydrogen release set up design was tested by placing an inverted glass funnel connected to a graduated burette placed directly above the sample to capture the hydrogen. Specimens were immersed in PBS, pH = 7.2, in a beaker placed in a thermostatic bath at a constant temperature 37 °C ± 1. The burette was filled with 10 mL of PBS and hydrogen release was measured by the displacement of PBS level in the burette as H_2_ gas evolves. The solution level in each burette was measured every day for 28 days. The volume of PBS in the beaker was calculated refer to the recommended conditions for bioactivity tests proposed by Kokubo [73] using the following equation:(6)$$k=\frac{S_{a}}{Vs}\;$$
where *S_a_* and *V_s_* are respectively the apparent surface area of immersed specimen (mm^2^) and the volume of PBS (mL) and *k* is the immersion coefficient (*k* equal to 9 mm^2^/g in our case). Three specimens for each formulation were soaked in 43 mL of PBS. Data are presented as mean value ± standard deviation.

### 3.5. Hydrolytic Degradation Behaviour

SBF (Appendix A) is commonly used to evaluate the behaviour of bioactive materials. Indeed, its ionic concentration and pH (7.4) match with those of human blood plasma. The volume of SBF was calculated as described previously. Samples were soaked in vials containing 43 mL of SBF with 0.02 wt.% of sodium azide to avoid any bacterial contamination [74]. The test was performed in the oven at 37 °C ± 1 °C during 12 weeks. At predetermined periods (two weeks, one month, one and a half month, two months, two and a half months and three months) the samples were picked out from the SBF and rinsed several times with distilled water. Finally, the specimen surface was wiped off using a paper and dried under nitrogen before characterizations.

### 3.6. Characterizations

#### 3.6.1. H^1^ NMR

The successfulness of the PEO-*b*-PLLA copolymer synthesis was monitored by ^1^H NMR using a Bruker AMX-500 spectrometer at 500 MHz. The ^1^H NMR spectrum was obtained at r.t. in 0.6 mL of CDCl_3_ solvent with tetramethylsilane (TMS) as an internal standard.

#### 3.6.2. Water Contact Angle (WCA)

Experiments were carried out at r.t. with a Drop Shape Analyzer DSA 10 MK2 KRUSS by the sessile drop method with deionized water as a polar liquid. The values of contact angles were the average of at least five drops on different points of the samples and they were reported with the standard deviation.

#### 3.6.3. Dynamic Mechanical Thermal Analysis (DMTA)

Dynamic mechanical thermal analysis (DTMA) was performed using TA, model Q800, in dual cantilever bending mode. The temperature was raised from 0 °C to 120 °C at a heating rate of 2 °C/min, a frequency of 1 Hz and an amplitude of 20 µm.

#### 3.6.4. Gel Permeation Chromatography (GPC)

The molecular weights of the neat PLA and reinforced composites were determined by gel permeation chromatography (GPC) in CHCl_3_ (2 mg polymer/1 mL solvent) at 30 °C using an Agilent liquid chromatograph equipped with an Agilent degasser, an isocratic HPLC pump (flow rate = 1 mL/min). PS standard was used for the calibration.

#### 3.6.5. Differential Scanning Calorimetry (DSC)

The thermal behaviour was analysed by differential scanning calorimetry, DSC model TA Q2000. Experiments were performed under nitrogen atmosphere (50 mL/min), using around 10 mg of each sample. The experimental procedure is investigated at the temperature range of 0 °C to 200 °C: a first heating to 200 °C, following by a cooling to 0 °C, and finally a second heating to 200 °C, all three of them at a rate of 10 °C/min. The glass transition (T_g_), crystallization (T_c_) and melting temperature (T_m_) of the PLA and the composites were taken from the second heating curve in order to erase the previous thermal history. For the degradation test, the degree of crystallization ($\chi_{c}$) was calculated following equation: (7)$$\left(\%\right)\chi_{c}=\frac{\Delta H_{m\left(t\right)}-\Delta H_{c\left(t\right)}}{\Delta H_{m}{}^{0}\;}*100\;$$
where ${\Delta H}_{m\left(t\right)}$ and ${\Delta H}_{c\left(t\right)}$ are respectively the melting and cold crystallization enthalpies of samples at the time t of degradation, both of them were taken from the first heating curve. ${\Delta H}_{m}{}^{0}$ is the melting enthalpy of 100 % crystalline PLA (93 J/g) [42].

#### 3.6.6. Mass Variation

All the specimens were weighted before immersion (at *t*_0_) and after drying the samples at 60 °C under nitrogen (*t_d_*). The percentage of mass gain of loss, i.e., mass difference between the dried specimen at *t_d_*, *M_d_*(*t_d_*), and the initial specimen at *t*_0_, *M*_0_(*t*_0_), with respect to the initial weight, was calculated according to equation:(8)$$\left(\%\right)gain\;\mathrm{or}\;loss\;weight=\frac{M_{d}{}_{\left(td\right)}-M_{0}{}_{\left(t0\right)}}{M_{0}{}_{\left(t_{0}\right)}}*100$$

Data are presented as mean value ± standard deviation.

#### 3.6.7. pH Assessment

In order to control the pH value as indicator for acidic degradation products, SBF was measured every two weeks during the soaking time using an 827 pH lab Metrohm. Data are presented as mean value ± standard deviation.

#### 3.6.8. Inductively Coupled Plasma (ICP)

An inductively coupled plasma-optic emission spectroscopy (720-ES ICP-OES, Agilent technologies with axially viewing and simultaneous CCD detection) was used to quantify the amount of Mg, Ca and P at 278.142, 318.127 and 213.547 nm, respectively, and released into SBF during the immersion time via the ICP Expert TM software (version 2.0.4). All the analyses were performed 40 min after the spectrometer was turned on to achieve a stable plasma as well as constant and reproducible sample introduction. The experiments has been performed 3 times for each formulation (PLA/10 Mg and PLA/10 Copo/10 Mg) overtime (after 2, 4, 6 and 8 weeks of degradation in SBF).

#### 3.6.9. Scanning Electronic Microscopy (SEM) and Energy Dispersive X-ray (EDX) Analysis

The morphology and element mapping of the specimens were investigated by a Scanning Electronic Microscopy (SEM), Hitachi SU8020, equipped with Energy-Dispersive X-ray (EDX) analyzer, Thermo Scientific Noran 7. The images were observed before immersion and at different degradation times. Samples were soaked in liquid nitrogen then fractured to expose the internal structure for the SEM. During the degradation, analyses were carried out at the surface and the cryo-fractured cross-sections to assess the morphology (SEM) and the chemical composition (EDX). The images were collected at an accelerated voltage of 15 and 20 kV. Before testing, all the specimens were coated with carbon sputtering to provide good conductivity.

#### 3.6.10. X-ray Diffraction (XRD)

The phase composition of the composites was characterized using an X-ray diffraction (XRD, Panalytical Empyrean) with copper X-ray tube, operating at 40 kV and 40 mA at a scanning rate of 8°/min from 10° to 80° with a step size of 0.026° and a dwell time of 72 s per step.

#### 3.6.11. Compression Test

A universal testing machine INSTRON, model 4505, with a 100 KN load cell force was used at ambient condition to evaluate the mechanical properties of the composites. The samples were tested at a speed of 30 mm·min^−1^ (0.5 mm·s^−1^, relative strain rate of 0.125 s^−1^) without preloading. Accurate strain is measured thanks to a LVDT linear displacement sensors. Stress–strain data were computed from load-displacement measurements. The compressive modulus was determined based on the slope of the initial linear region of the stress–strain curve and the average for each composition (five discs were tested) was calculated. The real accurate dimensions of the specimens were measured before the test.

#### 3.6.12. Protein Adsorption

Protein adsorption assessments were carried out according to our previously published protocol [58]. All films were cut in 0.5 cm^2^ square-shaped and incubated with 200 μL of BSA, 2% and 10% of FBS, and 5 mg/mL of plasma. Control experiments were carried out by incubating 0.5 cm^2^ square of each material with H_2_Od (deionized water). Proteins were incubated for two different time points: 30 min and 24 h at 37 °C. Total protein content, after three washing steps in H_2_Od, was measured by the Bradford method [75]. The absorbance at 595 nm was measured using a microtiter plate reader (ELISA reader, DV990BV6, GDV, Roma, Italy). Each sample was examined in 3 independent experiments, each run-in triplicate. Data reported are the mean value ± SD. *p* ≤ 0.05 was considered statistically significant.

#### 3.6.13. Human Adult Mesenchymal Stem Cells In Vitro Culture

Human bone marrow mesenchymal stem cells (hBM-MSCs) and human adipose stem cells (hASCs) were cultured according to our previous works [54,58,60,68]. Both types of mesenchymal stem cells were isolated by our group from waste samples of donors’ surgery with informed consent in accordance with the Declaration of Helsinki. After isolation, mesenchymal stem cells phenotype was analysed by measuring the expression of surface markers CD73, CD90, CD45, and CD105 (BD Biosciences, San Jose, CA, USA) using the flow cytometer FACScan (BD Biosciences, San Jose, CA, USA) and the FlowJo software (Tree Star, Ashland, OR, USA version 10.0.1, accessed on 2012) as described in previous works [54,64,65]. hBM-MSCs were cultured in DMEM High Glucose with 10% FBS, 1% penicillin–streptomycin and 2 mM L-glutamine in a humidified atmosphere and 5% of CO_2_ at 37 °C. hASCs were cultured in RPMI-1640 supplemented with 10% FBS, 1% penicillin-streptomycin and 1% L-glutamine at 37 °C, 5% CO_2_. Finally, the medium was changed every three days in both mesenchymal stem cell cultures.

The multipotential properties of both stem cell types were evaluated by inducing the adipogenic and osteogenic differentiation capability [50,54,56,64].

The adipogenic differentiation was performed by using the Lonza’s hMSC Adipogenic Differentiation BulletKit™ Medium (Catalog #: PT-3004, Lonza, Walkersville, Inc., Walkersville, MD, USA). The adipogenic differentiation was evaluated by Oil Red O Staining Kit (ScienCell Research Laboratories, Carlsbad, CA, USA), following the manufacture’s instruction. The staining was evaluated by the brightfield microscopy (Eclipse-TS100, Nikon, Tokyo, Japan) equipped with a digital SIGHT DS-5M-L1 photo camera (Nikon, Tokyo, Japan).

The osteogenic differentiation was performed by using the Lonza’s hMSC Osteogenic Differentiation BulletKit™ Medium (Catalog #: PT-3002, Lonza, Walkersville, Inc., Walkersville, MD, USA). The osteogenic differentiation was evaluated by Alizarin Red S Staining Kit (ScienCell Research Laboratories, Carlsbad, CA, USA) following the manufacture’s instruction and photo were captured with the brightfield microscopy (Eclipse-TS100, Nikon, Tokyo, Japan) equipped with a digital SIGHT DS-5M-L1 photo camera (Nikon, Tokyo, Japan).

#### 3.6.14. Culture of Human Adult Mesenchymal Stem Cells on PLA-Based Films

Each PLA-based film was cut in 1 cm^2^ square-shape and sterilized thanks to the immersion in 70% ethanol (for 30 s), rinsed with sterile PBS, and finally placed in multi-well plates. Once the film was dried, stem cells suspension was seeded dropwise on the film surfaces. Culture medium, after 45 min, was gently added to each film and stem cells/PLA-based film cultures were incubated under canonical culture conditions (humidified atmosphere at 37 °C, 5% CO_2_). Every three days the medium was changed, and stem cell cultures were analysed for viability and morphology at different time points.

##### Stem Cell Proliferation on PLA-Based Films

Stem cells proliferation was performed by seeding hBM-MSCs and hASCs on PLA-based films and on tissue culture polystyrene (TCP) as internal control (CTR) in growth medium at a starting concentration of 1.5 × 10^3^ cells/mL. Cells were counted at 3, 7, 14D. After trypsinization the stem cell pellet was re-suspended in PBS, then the cell aliquot was mixed with Trypan Blue solution, and counted by the Invitrogen™ Counteness™ automated cell counter (Invitrogen™, Grand Island, NY, USA).

##### Stem Cell Viability on PLA-Based Films

Viability of stem cells was evaluated by seeding hBM-MSCs and hASCs on PLA-based films and on TCP as control (CTR), in growth medium at a starting concentration of 1.5 × 10^3^ cells/mL. Cell viability was evaluated by using the mitochondrial dehydrogenase activity incubating cell cultures with MTT (3-(4,5-dimethylthiazol-2-yl)-2,5-diphenyltetrazolium bromide) (Sigma Aldrich, St. Louis, MI, USA) according to the manufacturer’s recommendations. The absorbance was measured at 589 nm with a reference wavelength at 650 nm using a microtiter plate reader (ELISA reader, DV990BV6, GDV, Roma, Italy). At different times (3, 7 and 14 days (D)), cell viability was evaluated. Every experiment was performed in triplicate.

##### Immunofluorescence

After sterilization of PLA-based films, stem cell suspensions of 1.5 × 10^3^ of hBM-MSCs and hASCs were seeded on each film surface, and then 500 μL of culture medium was gradually added. As a CTR experiments were performed seeding stem cells on glass coverslip (GC). The morphology of stem cells was analysed by immunostaining of the cytoskeleton at different time points (3D, 7D and 14D). Briefly, the stem cells on different films were rinsed twice with PBS, fixed in 4% paraformaldehyde for 20 min, rinsing with PBS, permeabilized (PBS + 3% FBS + 0.5% Triton X-100) and finally blocked (PBS + 3% FBS + 0.05% Triton X-100) for 1 h at r.t. To achieve F-actin fibres, samples were incubated with phalloidin (Alexa-fluor-488 phalloidin, Invitrogen, Grand Island, NY, USA) for 20 min at r.t. and then after washing with PBS, samples were mounted and nuclei were counterstained with Vectashield^®^ with DAPI. Image acquisition was performed by using a fluorescence microscope (Eclipse-TE2000-S, Nikon, Tokyo, Japan) equipped with the F-ViewII FireWire camera (cell Soft Imaging System, Olympus, Germany, version 2.5, Accessed in 2006). PLA films without cells were also evaluated to analyse their interference.

#### 3.6.15. Statistical Analysis

Experimental data were reported as the mean ± SD (GraphPad Software, Inc., San Diego, CA, USA, version 4.0). Post-hoc comparison test was carried out using the ANOVA (one-way analysis of variance) and Dunn’s multiple comparison test (GraphPad Software, Inc., La Jolla, CA, USA, version 4.0, Accessed in 2008). *p* ≤ 0.05 was considered statistically significant.

## 4. Conclusions

The current study aimed on evidencing the positive role of the amphiphilic diblock PEO-*b*-PLLA copolymer as a novel interface, improving the interactions between the metallic hydrophilic Mg filler and the hydrophobic PLA matrix. This was achieved by studying the influence of the copolymer incorporation on the PLA/Mg surface stabilization, hydrophilicity, degradation, bioactivity and stem cells behaviour. The principal findings were as follows:The affinity between the matrix and fillers was enhanced. Therefore, an increase of the PLA/Mg interfacial adhesion was observed in SEM analysis with the different amounts of fillers. This behaviour was also confirmed by a decrease in the damping factor.A decrease in the WCA measurements was shown. Indeed, an increase of the composite hydrophilicity by selective surface localization of the hydrophilic PEO present in the surfactant was proven. This behaviour has a significant effect on the protein adsorption, which is an interesting finding.During the degradation in SBF, hydroxyapatite, the major component of natural bone, has been formed in bulk after 8 weeks of immersion, which could induce good biocompatibility and osteoconductivity.Viability test showed that the PLA, PLA/10 Mg and PLA/10 Copo/10 Mg were suitable for mesenchymal stem cell cultures. Moreover, cell morphology observations confirmed a maintenance of the monolayer fibroblast-like morphology. Therefore, the stem cells are still alive following the copolymer addition. This finding showed that these films could serve as implants for biomedical applications.However, a slight decrease in the mechanical was revealed due to the PEO plasticizing effect. In this regard, further work is in progress the optimize these promising PLA/Mg bioresorbable design by preparing novel co(polymer).

## Figures and Tables

**Figure 1 molecules-26-05944-f001:**
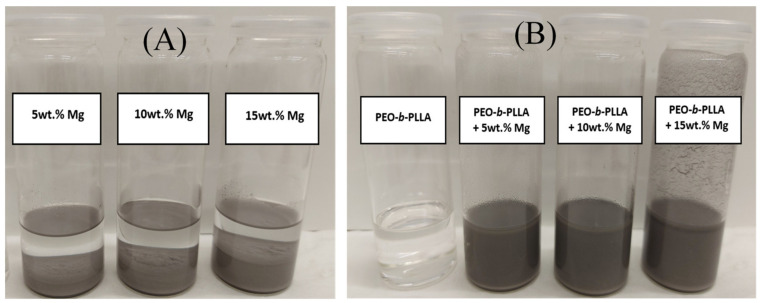
Dispersion tests with 5, 10 and 15 wt.% of Mg filler in CHCl_3_: (**A**) in absence and (**B**) with 10 wt.% of PEO-b-PLLA copolymer after 4 h.

**Figure 2 molecules-26-05944-f002:**
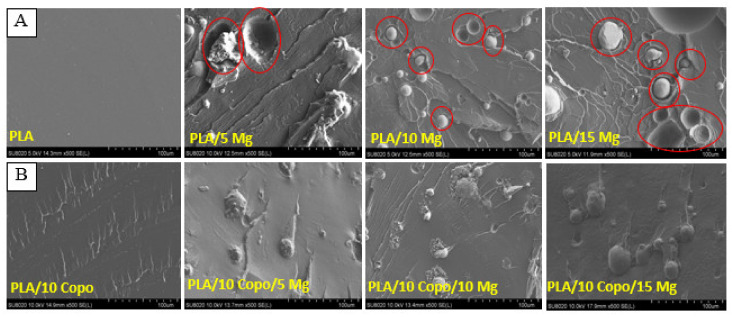
SEM images of cryofractured (**A**) PLA/x Mg and (**B**) PLA/10 Copo/x Mg composites with an Mg amount of x = 0, 5, 10 and 15 wt.%.

**Figure 3 molecules-26-05944-f003:**
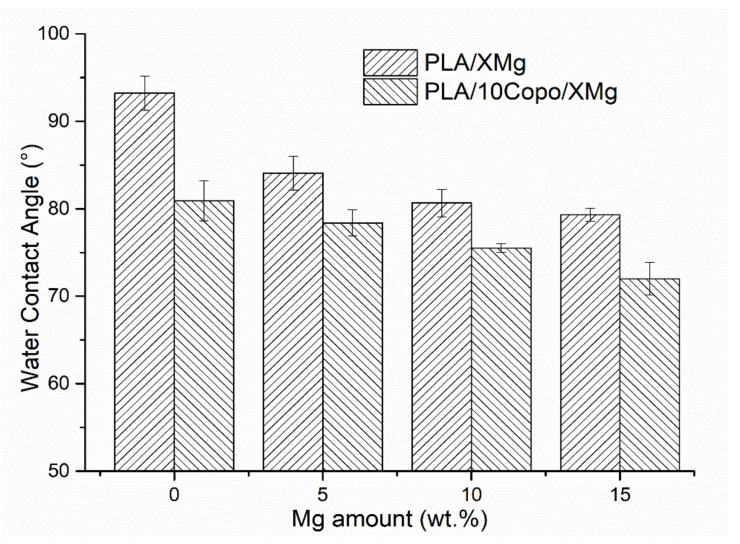
Water Contact Angle of PLA/x Mg and PLA/10 Copo/x Mg composite surfaces versus Mg amount (x = 0, 5, 10 and 15 wt.%).

**Figure 4 molecules-26-05944-f004:**
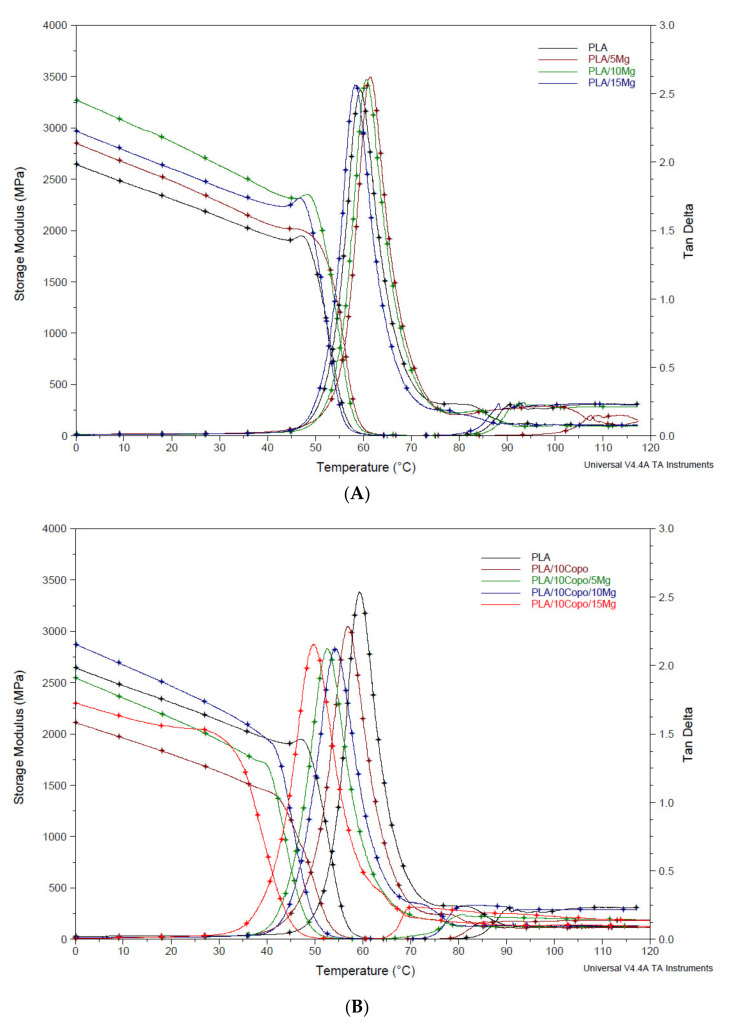
DMTA curves of (**A**) PLA/x Mg and (**B**) PLA/10 Copo/x Mg composites (x = 0, 5, 10 and 15 wt.%).

**Figure 5 molecules-26-05944-f005:**
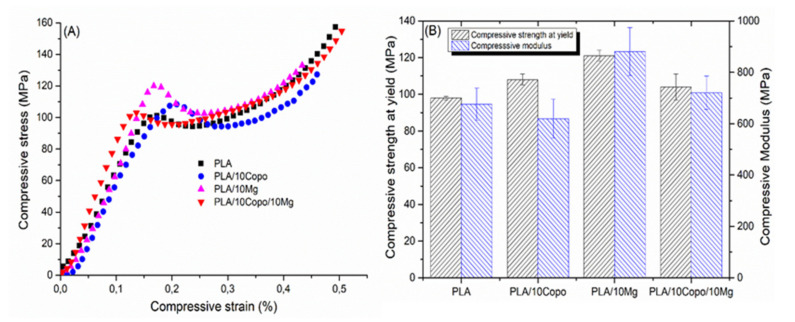
(**A**) Compressive stress vs. strain curves and (**B**) compressive strength at yield and compressive modulus of the composites.

**Figure 6 molecules-26-05944-f006:**
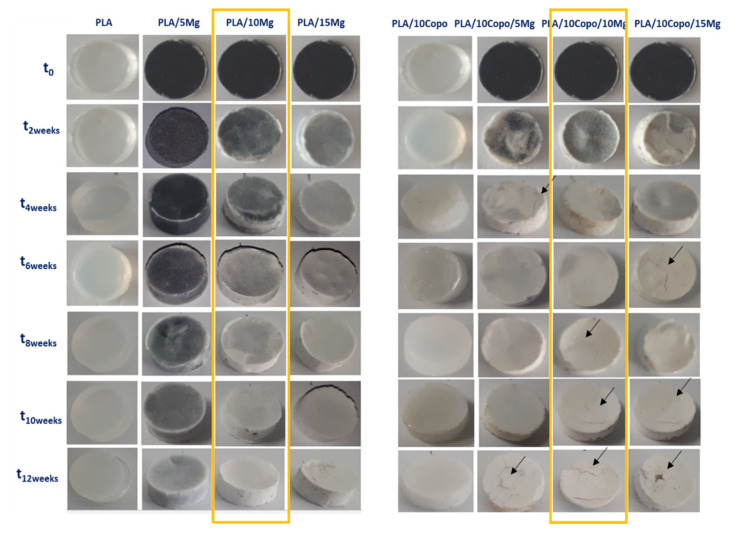
Visual evolution of all the specimens during the degradation time at 37 °C in the SBF.

**Figure 7 molecules-26-05944-f007:**
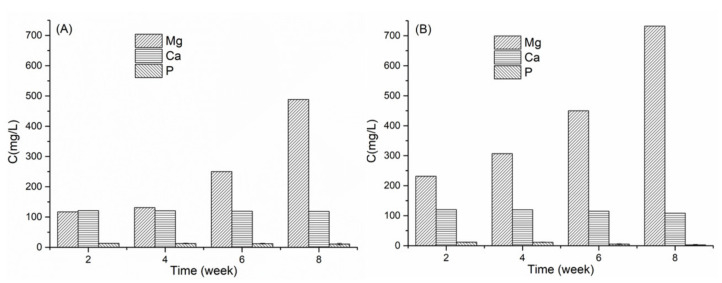
Ions concentration changes (Mg, Ca, P) during the degradation process in SBF (**A**) PLA/10 Mg and (**B**) PLA/10 Copo/10 Mg composites.

**Figure 8 molecules-26-05944-f008:**
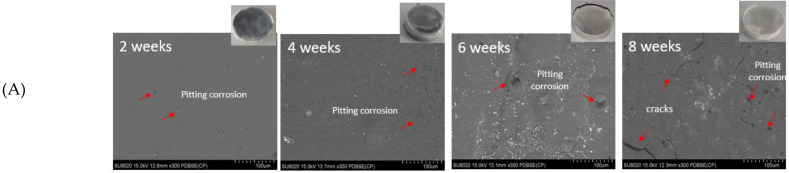
Surface morphology: SEM images of (**A**–**C**) PLA/10 Mg and (**B**–**D**) PLA/10 Copo/10 Mg composites during the immersion in SBF.

**Figure 9 molecules-26-05944-f009:**
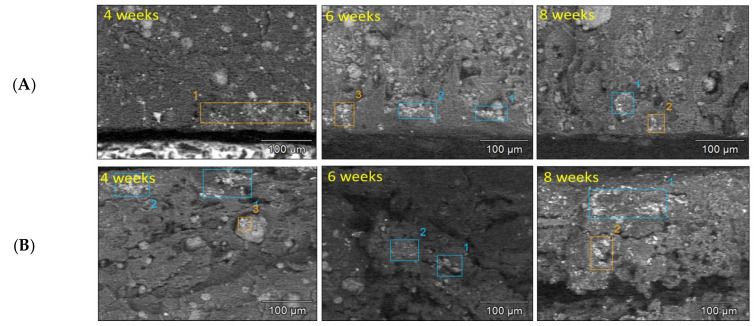
Cross-sectional morphology: SEM images of (**A**) PLA/10 Mg and (**B**) PLA/10 Copo/10 Mg during the immersion in SBF.

**Figure 10 molecules-26-05944-f010:**
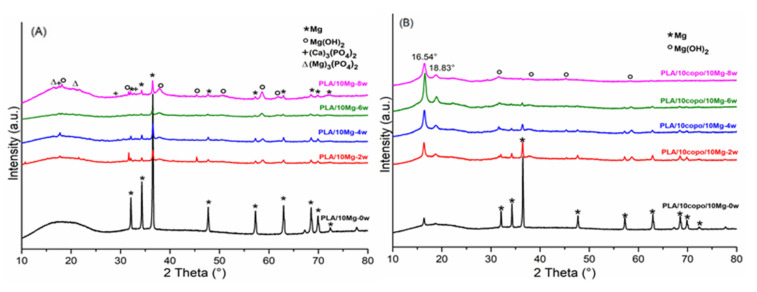
XRD patterns of (**A**) PLA/10 Mg and (**B**) PLA/10 Copo/10 Mg during the degradation process.

**Figure 11 molecules-26-05944-f011:**
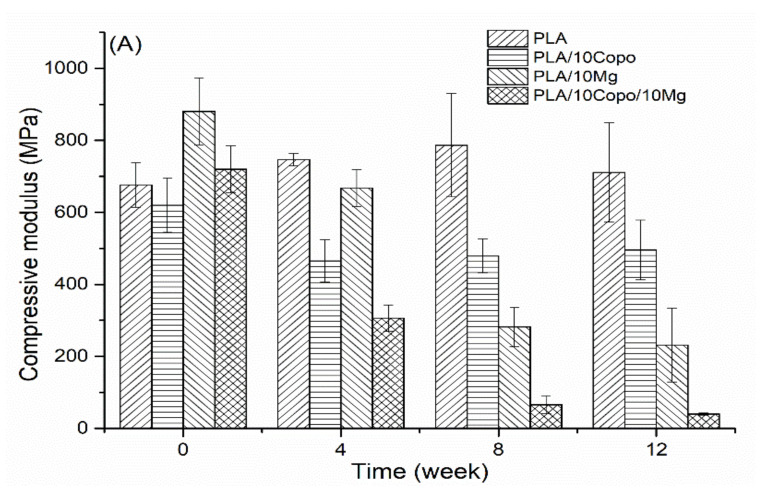
Mechanical behaviour: (**A**) Compressive modulus, (**B**) compressive strength at yield as a function of immersion time.

**Figure 12 molecules-26-05944-f012:**
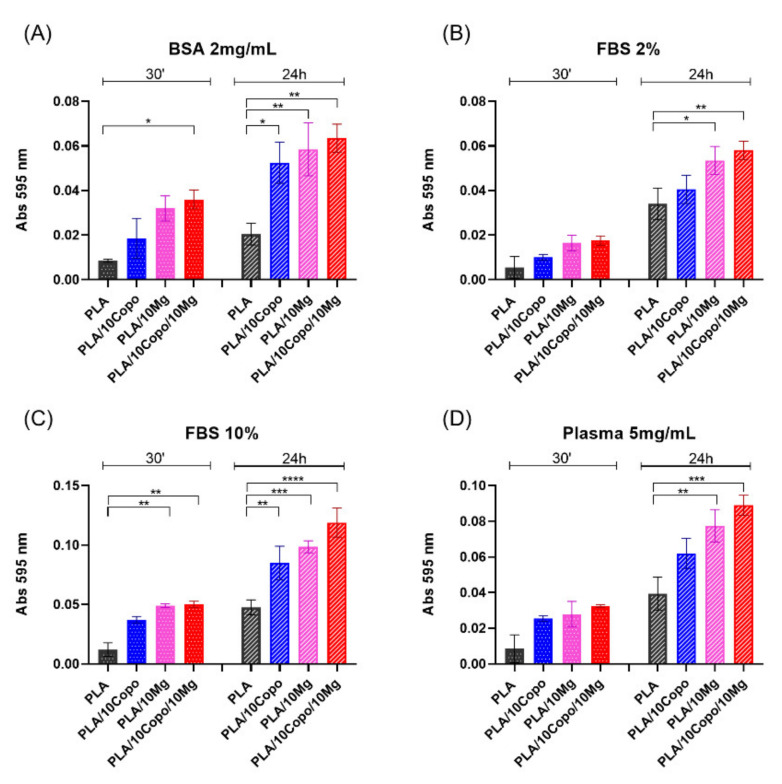
Adsorbed protein on PLA, PLA/10 Copo, PLA/10 Mg and PLA/10 Copo/10 Mg films. Total adsorbed proteins from (**A**) BSA 2 mg/mL, (**B**) plasma 5 mg/mL, (**C**) 2% of FBS and (**D**) 10% of FBS after 30 min and 24 h at 37 °C. Data are representative of three independent experiments and are reported in percentage with respect to the PLA as mean ± SD. Significance of the differences was indicated as follows: * *p* < 0.05; ** *p* < 0.01; *** *p* < 0.001; **** *p* < 0.0001.

**Figure 13 molecules-26-05944-f013:**
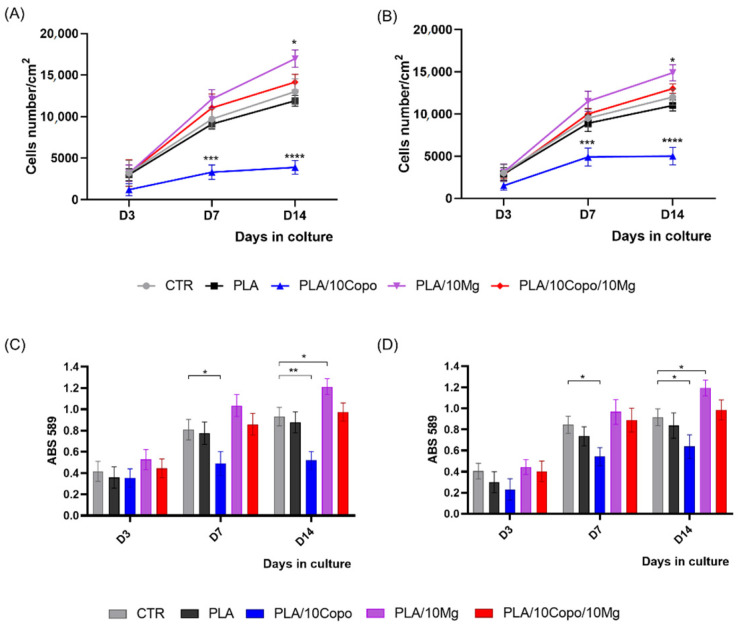
hMSCs proliferation (**A**,**B**) and viability (**C**,**D**). (**A**,**C**) hBM-MSCs and (**B**,**D**) hASCs on PLA, PLA/10 Copo, PLA/10 Mg and PLA/10 Copo/10 Mg and on canonical stem cells culture condition (CTR) at different time points D3, D7 and D14. Data are representative of three independent experiments and are reported as mean ± SD. Statistical analysis was made comparing cells viability on each PLA-based film with respect to the CTR. Significance of the differences was indicated as follows: * *p* <0.05; ** *p* <0.01; *** *p* < 0.001; **** *p* < 0.0001.

**Figure 14 molecules-26-05944-f014:**
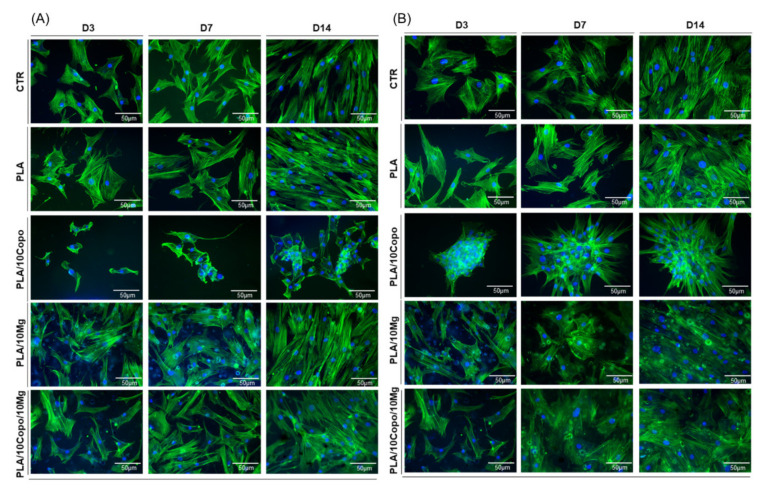
Morphology of (**A**) hBM-MSCs and (**B**) hASCs on PLA, PLA/10 Copo, PLA/10 Mg and PLA/10 Copo/10 Mg during the time in culture (D3, D7 and D14). Representative images of F-Actin (Phalloidin-Alexa-fluor-488) and nuclei (DAPI, blue) on PLA-based films and on control. Scale bar = 50 μm.

**Table 1 molecules-26-05944-t001:** Sample compositions after melt-blending.

Samples	Mg (wt.%)	PEO-b-PLLA (wt.%)	PLA (wt.%)
PLA			100
PLA/5 Mg	5		95
PLA/10 Mg	10		90
PLA/15 Mg	15		85
PLA/10 Copo		10	90
PLA/10 Copo/5 Mg	5	10	85
PLA/10 Copo/10 Mg	10	10	80
PLA/10 Copo/15 Mg	15	10	75

**Table 2 molecules-26-05944-t002:** Chemical composition: EDX analysis of degradation products formed on the surface of (A) PLA/10 Mg and (B) PLA/10 Copo/10 Mg composites during the immersion in SBF.

(A) Elem. Wt.%	O	Mg	Ca	P	Y *	Ca/P
PLA/10 Mg-4w	40.60	2.00	5.40	4.90	47.10	1.10
PLA/10 Mg-6w	45.42	1.60	3.00	1.90	48.10	1.57
PLA/10 Mg-8w	49.98	0.39	0.84	0.50	48.29	1.67
**(B) Elem. Wt.%**	**O**	**Mg**	**Ca**	**P**	**Y ***	**Ca/P**
PLA/10 Copo/10 Mg-4w	47.40	3.20	4.00	7.70	37.70	0.51
PLA/10 Copo/10 Mg-6w	53.70	0.70	0.50	1.60	43.50	0.31
PLA/10 Copo/10 Mg-8w	60.60	0.50	0.70	0.70	37.50	1.00

Y *: Carbone and salts.

**Table 3 molecules-26-05944-t003:** Chemical composition: EDX analysis of degradation products formed in bulk of (**A**) PLA/10 Mg and (**B**) PLA/10 Copo/10 Mg composites during the immersion in SBF.

(A) Elem. Wt.%	O	Mg	Ca	P	Y *	Ca/P
PLA/10 Mg-4w	45.70	3.69	6.51	-	44.10	-
PLA/10 Mg-6w	41.60	7.28	17.40	-	33.72	-
PLA/10 Mg-8w	56.10	3.51	1.50	1.50	37.39	1.00
**(B) Elem. Wt.%**	**O**	**Mg**	**Ca**	**P**	**Y ***	**Ca/P**
PLA/10 Copo/10 Mg-4w	47.20	13.60	8.50	-	30.70	-
PLA/10 Copo/10 Mg-6w	33.50	5.69	10.31	7.61	42.89	1.35
PLA/10 Copo/10 Mg-8w	50.79	3.29	5.68	3.40	36.84	1.67

Y *: Carbone and salts.

## Data Availability

The data presented in this study are available on request from the corresponding author.

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
