# Peer review of "Interfacial Compatibilization into PLA/Mg Composites for Improved In Vitro Bioactivity and Stem Cell Adhesion"

_molecules, 2021, doi:10.3390/molecules26195944_

Round 1

Reviewer 1 Report

I read with interest this well organised work on PLA/Mg composite compatibilization for bioactivity and cell adhesion. I would like to congratulate the authors for this comprehensive analysis. Some minor issues need to be addressed prior to publication:

All figure references have been expressed as "Error! Reference source not found" The authors need to provide the correct reference list.

Lines 504-507: i disagree with this statement. Other MSC types are also well characterised, easily isolated and well-differentiating. Please amend.

MTT is an assay that indicates metabolic activity. To assume that it represents viability the test needs to be applied to groups of comparable composition. This is tricky when comparing different biomaterials, since several papers have shown that metabolic activity of stem cells is affected by biomaterial composition (see Klontzas et al. Acta Biomaterialia 2019). Please mention this limitation and discuss the fact that metabolic activity is affected by the type of Biomaterials. To be able to demonstrate viability the numbers presented from your MTT experiments need to be normalised to cell numbers counted by means of protein or DNA. otherwise we do not know if a high value represents more cells or just more active  metabolism in the same number of cells. 

Calcein AM/EthD testing can provide more accurate results on the viability of your cells. 

Flow cytometric analysis of surface phenotype is not sufficient to call stem cells MSCs. The cells need to be tested for tri-lineage differentiation capacity according to the guidelines of the ISCT. Did you assess the differentiation capacity of your cells? if yes please provide flow cytometry and differentiation results of your cell batches.

No statistical analysis has been provided for any of the graphs apart from the stem cell ones. Please add. 

Figure 7 does not have any error bars. Did the authors perform the experiments more than once?

Using SEM as an error metric can be deceiving in such experiments because it does not represent the error in the whole population. Please present results using SD instead.

Author Response

First of all, Authors would like to thank the Editor and the Reviewer for their time and concern. Here below, we enclose our point-to-point answers to Reviewers’ comments.

Reviewer 1:

Recommendation: I read with interest this well organised work on PLA/Mg composite compatibilization for bioactivity and cell adhesion. I would like to congratulate the authors for this comprehensive analysis. Some minor issues need to be addressed prior to publicationminor issues need to be addressed prior to publication.

Comments:

  1. All figure references have been expressed as "Error! Reference source not found" The authors need to provide the correct reference list.

Response: Authors agree and thank the reviewer for this comment. Indeed, in the PDF version, there was a confuse to identify properly the figures in the appropriate paragraphs as they have been expressed as ‘Error’. To avoid any confuse, the formatting has been removed from all the figures to allow better visibility when reading the manuscript. This modification is now done for all the paragraphs.

  1. Lines 504-507: I disagree with this statement. Other MSC types are also well characterised, easily isolated and well-differentiating. Please amend.

Response: Authors agree and thank the reviewer for this comment. As requested, the statement was deleted and, additionally, the text revised.

  1. MTT is an assay that indicates metabolic activity. To assume that it represents viability the test needs to be applied to groups of comparable composition. This is tricky when comparing different biomaterials, since several papers have shown that metabolic activity of stem cells is affected by biomaterial composition (see Klontzas et al. Acta Biomaterialia 2019). Please mention this limitation and discuss the fact that metabolic activity is affected by the type of Biomaterials. To be able to demonstrate viability the numbers presented from your MTT experiments need to be normalised to cell numbers counted by means of protein or DNA. otherwise we do not know if a high value represents more cells or just more active metabolism in the same number of cells.

               Calcein AM/EthD testing can provide more accurate results on the viability of your cells.

Response: The authors thank the reviewer for this observation. We agree that the type of biomaterial influence the stem cell behaviuor. We have been investigating this issue in our several previous works (e.g.: D'Angelo F, Armentano I, Cacciotti I, Tiribuzi R, Quattrocelli M, Del Gaudio C, Fortunati E, Saino E, Caraffa A, Cerulli GG, Visai L, Kenny JM, Sampaolesi M, Bianco A, Martino S, Orlacchio A. Tuning multi/pluri-potent stem cell fate by electrospun poly(L-lactic acid)-calcium-deficient hydroxyapatite nanocomposite mats. Biomacromolecules. 2012 May 14;13(5):1350-60. doi: 10.1021/bm3000716.

Morena, F.; Armentano, I.; Montanucci, P.; Argentati, C.; Fortunati, E.; Montesano, S.; Bicchi, I.; Pescara, T.; Pennoni, I.; Mattioli, S.; Torre, L.; Latterini, L. ; Emiliani, C. ; Basta, G. ; Calafiore, R. ; Kenny, J.M., Martino, S. Design of a nanocomposite substrate inducing adult stem cell assembly and progression toward an Epiblast-like or Primitive Endoderm-like phenotype via mechanotransduction. Biomaterials 2017, 144, 211–229, doi:10.1016/j.biomaterials.2017.08.015.

Argentati, C.; Morena, F.; Montanucci, P.; Rallini, M.; Basta, G.; Calabrese, N.; Calafiore, R.; Cordellini, M.; Emiliani, C.; Armentano, I.; et al. Surface hydrophilicity of poly(L-lactide) acid polymer film changes the human adult adipose stem cell architecture. Polymers (Basel). 2018, 10, 1–17, doi:10.3390/polym10020140;

Morena, F.; Argentati, C.; Soccio, M.; Bicchi, I.; Lotti, N.; Armentano, I.; Martino, S. Unpatterned Bioactive Poly(Butylene 1,4-Cyclohexanedicarboxylate)-Based Film Fast Induced Neuronal-Like Differentiation of Human Bone Marrow-Mesenchymal Stem Cells. Int. J. Mol. Sci. 2020, 21, 9274.)

We have improved the results section including proliferation data of both stem cells on PLA-derived films and control culture. Moreover, data of proliferation were correlated with data of MTT. We have revised Figure 13, results and method sections.

  1. Flow cytometric analysis of surface phenotype is not sufficient to call stem cells MSCs. The cells need to be tested for tri-lineage differentiation capacity according to the guidelines of the ISCT. Did you assess the differentiation capacity of your cells? if yes please provide flow cytometry and differentiation results of your cell batches.

Response: The authors thank the reviewer for this note. The mesenchymal stem cell characterization includes also the evaluation of the multipotential capability. This is part of our experimental procedures when we isolated mesenchymal stem cells.  We included in the supplementary file Figure.S6 previously omitted information (adipogenic and osteogenic differentiation) and the flow cytometric analysis of both stem cell types. We have also revised the method section.

  1. No statistical analysis has been provided for any of the graphs apart from the stem cell ones. Please add

Response: Authors agree principally. However, statistical analysis are not required for analytical methods where the method error does not exceed 10% and samples reproducibility is also good. In order to verify sample reproducibility, all tests have been performed on at least 3 different samples from the same composition. To avoid any further misleading, the number of tested samples/analysis is given in the Experimental part of the revised manuscript.

  1. Figure 7 does not have any error bars. Did the authors perform the experiments more than once?

Response: Authors agree totally with the reviewer. Indeed, it is a bit tricky to see the error bars of figure 7. However, the experiments has been performed 3 times for each formulation. In other word, the concentration of Mg, Ca and P in the SBF was analyzed 3 times both for PLA/10Mg and PLA/10Copo/10Mg overtime (after 2, 4, 6 and 8 weeks of degradation). Unfortunately, error bars are not visible as the range or error is minimal compared to Mg, Ca and P concentration. A describing sentence is added to the revised manuscript, in the materials and methods section (3.6.8. Inductively coupled plasma) to avoid any misleading and misunderstanding. Another sentence has been also added to the figure S2 title in the supporting information file.

For more clarification, here is the error bars ranges overtime for each component in:

PLA/10Mg composite: Mg [0.113 - 2.766] mg/L; Ca [0.288 - 0.969] mg/L; P [0.040 - 0.058] mg/L.

PLA/10Copo/10Mg composite: Mg [0.683 - 1.844] mg/L; Ca [0.172 - 0.473] mg/L; P [0.005 - 0.038] mg/L.

  1. Using SEM as an error metric can be deceiving in such experiments because it does not represent the error in the whole population. Please present results using SD instead.

Response: As requested we have revised the results section. Now data are reported as the mean ± SD.

Reviewer 2 Report

The article is well designed and is recommended to be accepted after minor modifications below:

Please refer to other materials used in this field in the introduction, such as the following references:

https://doi.org/10.1016/j.matchemphys.2019.122305

Author Response

First of all, Authors would like to thank the Editor and the Reviewer for their time and concern. Here below, we enclose our point-to-point answers to Reviewers’ comments.

Reviewer 2:

Recommendation: The article is well designed and is recommended to be accepted after minor modifications.

Comments:

  1. Please refer to other materials used in this field in the introduction, such as the following references: https://doi.org/10.1016/j.matchemphys.2019.122305

Response: Authors would like to thank the reviewer for this interesting reference. Indeed, the combination between hydroxyapatite and graphene increase the mechanical properties of the scaffold. The refence was added to the introduction (reference 13) as recommended.

Reviewer 3 Report

The paper may be of interest for the journal. It descrive the characterization of PLA/Mg composite scaffolds for bone regeneration procedures.

I suggest major changes of the paper.

 IN the introduction, several recent studies included bioactive fillers (CaSi-DCPD) widely used in dentistry and endodontic field to overcome the release of acidic products and the lack of wettability of PLA  with the development of highly porous structures. This could be mentioned as another approach in the introduction (Page 2 line 58).

Several studies investigated newly designed polymeric scaffolds (highly porous casi-dcpd doped poly (α-hydroxy) scaffolds, Mineral-doped poly(L-lactide) acid scaffolds enriched with exosomes, PLA-based mineral-doped scaffolds)  for bone regeneration. The characterization of these materials demonstrated apatite nucleating ability, release of biologically relevant ions, high porosity, and differentiation of cultured MSC (VW-MSC, hADMSC, hPC-MSC) into osteogenic or angiogenic lineage.
The authors should include and discuss some of the recent studies to evidence the importance of scaffold based approach in the field of tissue enginnering. This could be done in the introduction.

Materials and methods s missing, please provide the paragraph with, polymer synthesis, Mg fillers inclusion and tests performed in this chapter! All studies need a materials and method section in order to assess the methodology used and allowing to replicate the exact conditions.

Now I see that MeM chapter is after the results section. Please move MeM before as chapter 2…..

The test of pH of scaffolds immersed in SBF is not standard. Usually it is assessed in deionized water. Explain the rationale why you used this methodology (or a ISO).

Conclusions should be synthetized, to allow the reader to rapidly understand the key point of the research.

Author Response

First of all, Authors would like to thank the Editor and the Reviewer for their time and concern. Here below, we enclose our point-to-point answers to Reviewer´s comments.

Reviewer 3:

Recommendation: The paper may be of interest for the journal. It describes the characterization of PLA/Mg composite scaffolds for bone regeneration procedures.

I suggest major changes of the paper.

Comments:

  1. In the introduction, several recent studies included bioactive fillers (CaSi-DCPD) widely used in dentistry and endodontic field to overcome the release of acidic products and the lack of wettability of PLA with the development of highly porous structures. This could be mentioned as another approach in the introduction (Page 2 line 58).

Response: Authors agree with the reviewer. Indeed, in order to show the promising effect of bioactive fillers such as calcium silicates (CaSi) and dicalcium phosphate dihydrate (DCPD) a new reference was added to the introduction (doi:10.3390/nano10020243).

  1. Several studies investigated newly designed polymeric scaffolds (highly porous casi-dcpd doped poly (α-hydroxy) scaffolds, Mineral-doped poly(L-lactide) acid scaffolds enriched with exosomes, PLA-based mineral-doped scaffolds)  for bone regeneration. The characterization of these materials demonstrated apatite nucleating ability, release of biologically relevant ions, high porosity, and differentiation of cultured MSC (VW-MSC, hADMSC, hPC-MSC) into osteogenic or angiogenic lineage.
    The authors should include and discuss some of the recent studies to evidence the importance of scaffold based approach in the field of tissue engineering. This could be done in the introduction.

Response: Authors would like to thank to reviewer for this comment. As a response, recent studies have been discussed in the revised manuscript (introduction section) to evidence the importance of scaffold based approach in the bone regeneration applications.

  1. Materials and methods s missing, please provide the paragraph with, polymer synthesis, Mg fillers inclusion and tests performed in this chapter! All studies need a materials and method section in order to assess the methodology used and allowing to replicate the exact conditions.

Now I see that MeM chapter is after the results section. Please move MeM before as chapter 2…..

Response: Authors agree principally but we followed the template of Molecules journal.

  1. The test of pH of scaffolds immersed in SBF is not standard. Usually it is assessed in deionized water. Explain the rationale why you used this methodology (or a ISO).

Response: Authors agree that usually biological assessments need to follow standards. However; in some cases standards deviate from reality. To perform evaluations in conditions as close as possible to body medium, simulated body fluid (SBF) is used in this study. Indeed, its ion composition and concentration is very close to human blood plasma and thus allows having more correct picture of processes that might happen after material implantation. Authors would like to point that the used protocol is also applied by ther groups, cited below. More information about it can be found following the link (http://www.life.kyutech.ac.jp/~tmiya/SBF-e.html). In our study 0.02 wt.% of sodium azide was added to avoid any bacterial contamination (described in section 3. Matrials and methods, 3.5. Hydrolytic degradation behavior).

The ion concentration of the SBF is also presented in table S3 (supporting information file).

* Some degradation test for biomedical applications performed in SBF medium:

https://doi.org/10.1016/j.msec.2016.08.051

https://doi.org/10.1016/j.compscitech.2017.04.037

https://doi.org/10.1016/j.biomaterials.2006.01.017

DOI:10.1021/bm101327r

  1. Conclusions should be synthetized, to allow the reader to rapidly understand the key point of the research.

Response: Authors fully agree with this comment. The conclusions of the revised manuscript were therefore modified according to the Reviewer´s comment. 

Round 2

Reviewer 3 Report

the manuscript significantly improved.

the authors answered satisfactorialy and paper can be accepted